# Bag of Tricks for Improving Deep Learning Performance on Multimodal Image Classification

**DOI:** 10.3390/bioengineering9070312

**Published:** 2022-07-13

**Authors:** Steve A. Adeshina, Adeyinka P. Adedigba

**Affiliations:** 1Department of Computer Engineering, Nile University of Nigeria, Abuja 900001, Nigeria; 2Department of Mechatronics Engineering, Federal University of Technology, Minna 920211, Nigeria

**Keywords:** bag of tricks, COVID-19, label smoothing, lookahead optimizer, medical images, multi-modality, self-attention

## Abstract

A comprehensive medical image-based diagnosis is usually performed across various image modalities before passing a final decision; hence, designing a deep learning model that can use any medical image modality to diagnose a particular disease is of great interest. The available methods are multi-staged, with many computational bottlenecks in between. This paper presents an improved end-to-end method of multimodal image classification using deep learning models. We present top research methods developed over the years to improve models trained from scratch and transfer learning approaches. We show that when fully trained, a model can first implicitly discriminate the imaging modality and then diagnose the relevant disease. Our developed models were applied to COVID-19 classification from chest X-ray, CT scan, and lung ultrasound image modalities. The model that achieved the highest accuracy correctly maps all input images to their respective modality, then classifies the disease achieving overall 91.07% accuracy.

## 1. Introduction

Medical imaging provides different modalities for examining a particular disease in the body. The available imaging mode includes ultrasound, x-ray imaging, computer tomography (CT), magnetic resonance imaging (MRI), and positron emission tomography (PET), which provide a distinct representation of the internal organs for diagnosis purposes. What is obscure in one particular imaging modality can be easily observed using another modality; hence, multimodal image analysis is a common technique across the field of radiology [1]. However, some imaging modalities may be preferred to others based on cost, radiation dose, and risk [2]. In addition, due to the availability of expert radiologists, some imaging modes may be unavailable in some hospitals; this is usually the case in most developing countries.

Most research works on developing computer-aided techniques for diagnosing different kinds of diseases from medical images focus on a single image mode. Such models are irrelevant to hospitals that do not have that mode. Consequently, developing deep learning models that can accurately diagnose disease from different image modalities is of interest. Some researchers have risen to this challenge (see Section 2), but their approach includes the computational overhead of feature extraction and feature engineering. Therefore, this paper examines an end-to-end training approach, improved with different techniques for better performance.

We investigated various research works that address different issues related to training deep learning models, such as limited activation fields of convolution kernels, adversarial attacks due to overconfident class labels, death of neurons by ReLu activation function, and sensitivity of optimizers to hyperparameter selection (see Section 3.2 and Section 3.3 for details). Focusing on recent papers, we combined several of these techniques and designed experiments to observe their performance improvements (see Section 4). The results of these experiments are presented in Section 5.

This work is based on the hypothesis that a high-performance model would implicitly learn to classify input images based on their correct modality before it could correctly predict the correct class. In summary, the goal of this paper is to design a deep learning framework for diagnosing diseases from multimodal medical images, and the following are the contributions:Classification of COVID-19 from different imaging modalities such as chest x-ray (CXR), lung ultrasound (LUS), and CT scan using end-to-end training.Implementation of a bag of new tricks (such as a self-attention module with ResNet, new activation function, label smoothing, and *k*th—lookahead optimizer) for improving deep learning performance on multimodal classification.Improvement of deep learning models (trained from scratch and via transfer learning) using the bag of tricks.

In Section 2, we present a review of similar works, thereby establishing our contribution.

## 2. Review of Related Works

This section considers related works on multimodal classification, emphasizing the imaging modes considered and their framework. We begin with research efforts on COVID-19, then extend to other diseases.

Classification of COVID-19 from multimodal imaging using transfer learning is presented in [3]. The image modes considered are CXR, LUS, and CT scan. The author trained seven deep learning models: VGG, ResNet, Inception, Xception, InceptionResNet, NASNET, and DenseNet. However, in training the models, the author separated the image modes into a different batch of experiments, such that only one imaging mode is presented to the model at a time. Consequently, the model does not learn the imaging modality; instead, the problem is only a binary classification. In addition, the model’s performance to multimodal input cannot be tested; the result reported is for distinct image modality.

A different approach was taken by [4] in diagnosing COVID-19 from multimodal images. The author considered two modes: CT scan and CXR. Two different models (one for each image modality) were trained for feature extraction using transfer learning, then the features from these models were concatenated by another model. A similar method was used in [5] for classifying brain tumors. This approach has a drawback: the two image modes must be supplied simultaneously to get a result from the model; hence, when one mode is unavailable, the model is useless. In addition, this data fusion approach does not allow the model to discriminate image modes which could help enhance its inference process. A pre-trained DenseNet model was fine-tuned for multimodal brain tumor classification in [6]. The model was used for feature extraction; prominent features were selected using a modified genetic algorithm (MGA) and Entropy-Kurtosis-based High Feature Values (EKbHFV). The selected features were used to train a support vector machine for classification. The framework achieved 95% accuracy. However, this method does not provide end-to-end training, as the feature engineering layer (GA and EKbHFV) constitute a major computational bottleneck.

The image fusion method was used for multimodal skin lesion classification in [1]. A fusion of dermatoscopic and macroscopic images of the same lesion was fused with the metadata of the patient. First, 2 separate convolution neural networks were used for feature extraction and a neural network with 11 neurons for the metadata; then, the features were concatenated as input to another neural network. The framework achieved 72.1% accuracy and 72.6% precision.

In summary, the general framework for multimodal classification found in the literature is as follows: design a feature extractor for each image mode, concatenate the feature, and train a classifier. We note that this method is not end-to-end training and that several computational overheads are introduced because of this approach. The method proposed in this work is presented in the next section.

## 3. Methodology

A comprehensive medical image-based diagnosis is usually performed across various image modalities before passing a final decision, hence designing a deep learning model that can take any medical image mode to diagnose a particular disease. Accordingly, this paper explores the deep learning-based classification of multi-modal medical imaging to diagnose COVID-19. The medical image modalities considered in this work include chest X-ray (CXR), CT scan, and lung ultrasound (LUS).

We considered two approaches, as shown in Figure 1. Approach 1 considers a cascaded approach where a model classifies the image into its correct mode (LUS, CT scan, or CXR), then passes the image to the particular model designed for the modality. In Approach 2, a single model is trained to classify the image into COVID and non-COVID correctly.

### 3.1. Dataset

The dataset used for this work was gathered from different repositories as follows. Lung Ultrasound: This is an open-access benchmark data of COVID-19-related lung ultrasound image (called COVIDxUS) collected and curated by Ebadi et al. [7]. The dataset contains 174 videos of lung ultrasounds from COVID patients, non-COVID patients with reported lung infection, and normal LUS images for the control study. The dataset was gathered from six sources: the POCUS Atlas, GrepMed, the Butterfly Network, Life in the Fast Lane (*LITFL*), The Radiopaedia, and the CoreUltrasound. It should be noted that both linear and convex ultrasound probes were used for data collection.

However, the goal is to classify an LUS as COVID-19 or non-COVID-19; the videos of normal and others were merged to form the non-COVID class, while the COVID-19 row represents its class.

CT Scan: This dataset was collected from Hospitals in Sao Paulo, Brazil, and made publicly available by Angelov and Soares [8]. The dataset contains 2484 CT scan images, out of which 1252 were positive for COVID-19 while 1230 were negative.

Chest X-ray: This dataset contains the posterior–anterior (PA) and anterior–posterior (AP) views of the CXR, as radiologists commonly use these. The dataset was gathered and curated by Chowdhury et al. [9] from six different databases and is publicly available on Kaggle.

The summary and distribution of the datasets are presented in Table 1.

#### Data Pre-Processing and Augmentation

The only data pre-processing carried out in this work is in line with [10], where the LUS videos were converted to images, and histogram equalization was used for intensity normalization and scaling.

Similarly, the data augmentation was carried out to enhance our model and learn invariant features from the images. Therefore, various transformations were carried out on the dataset to obtain a different variant of the same image using the algorithm developed in our previous work [11]. The parameters of these transformations are presented in Table 2.

### 3.2. Modified CNN Architectures

The ResNet model by He et al. [12] forms the base model for this project. ResNet is the first deep learning model to surpass human-level performance in the 2015 ImageNet challenge with a 3.56% top-5 error rate. Since then, it has been an extensively studied network.

Several studies have shown that deeper networks perform substantially better than shallower counterparts. However, deeper networks are more prone to vanishing gradient problems, making them difficult to train. This problem was addressed by the implementation of the Residual block in ResNet. The Residual block modeled in Equation (1) creates a shortcut connection between the output of a convolutional layer and the earlier input to the layer using identity mapping [12]. Thus, the activation of a Residual block is given as:(1)$$x_{l}=ℱ\left(x_{l-1}\right)+x_{l-1},$$
where $x_{l}$ is the activation of layer l,ℱ(⋅) is a nonlinear convolutional transformation of the layer and $x_{l-1}$ is the activation of the previous layer l−1. The skip connection of Equation (1) enables more layers to be stacked on each other, resulting in a remarkably deep network.

#### 3.2.1. XResNet

ResNet architecture, as interpreted by He et al. [13], consists of the input stem, followed by four intermediate stages and the final output layer. The input stem reduces the image size by 4 times while increasing the number of channels to 64. This was done by applying a 7×7 convolution filter with a stride of 2 and an output channel of 64, followed by a 3×3 max-pool layer with a stride of 2.

The subsequent intermediate stages begin with a down-sampling block, followed by several residual blocks. The down-sampling block (shown in Figure 2a ) consists of a convolution path and a projection shortcut path. The convolution path has 3 convolutions whose filter sizes are 1×1, 3×3, and 1×1, respectively, all with a stride of 2, while the projection path has a convolution of filter size 1×1 with a stride of 2, used to match the input–output dimension.

In the PyTorch implementation of ResNet, it was observed that the convolution path ignores 3/4 of the input feature map because it uses a filter size of 1×1 with a stride of 2. Hence, PyTorch implements the convolution path with 2 1×1 filters with a stride of 1, while the 3×3 filter is implemented with a stride of 2.

Even so, Ref. [13] discovered that the PyTorch implementation also ignores 3/4 of the input feature map in its projection path due to the 1×1 filter size with a stride of 2. This was then replaced with a 1×1 convolution with a stride of 1 and a 2×2 average pooling layer with a stride of 2, as shown in Figure 2b. The ensuing model was called XResNet [13].

#### 3.2.2. Self-Attention Module

Understanding the convolution operation in deep CNN has shown that the convolutional layer only processes information in a local neighborhood defined by the receptive field of its filters (i.e., the filter size). To increase the receptive field of the convolution filter, the Residual network combines the feature maps of a filter with the original input feature to the layer in the residual module of ResNet. In DenseNet [14], the input feature to the dense block is connected to all the subsequent convolution layers in the dense block, and a convolution layer receives the activation maps from all its previous layer to ensure a wider receptive field.

Zhang et al. [15] assert that using the convolutional layer alone is computationally inefficient for long-range modeling dependencies in images. He proposed the self-attention (SA) module for generative adversarial networks, which enables both generator and discriminator to efficiently model relationships between widely separated spatial regions. The resulting network was called self-attention generative adversarial networks.

We replaced all convolution layers with the SA module in the base ResNet model in this work. The core similarity of the residual module and SA module is presented in Figure 3.

Like a convolution layer, the self-attention module estimates the response at a point as a weighted sum of all the features at that position by attending to all positions within the same region.

Given an input feature, x from the previous layer, the module first estimates the attention by transforming x into two different feature spaces Ψ and Ω, respectively, where:$$\Psi \left(x\right)=W_{\Psi }x\;\mathrm{and}\;\Omega \left(x\right)=W_{\Omega }x$$

The level of attention a model gives to the location i while synthesizing the
jth region is represented by softmax layer (
$\beta_{i,j}$) given by:
(2)$$\beta_{i,j}=\frac{\exp\left(\gamma_{i,j}\right)}{\sum_{i=1}^{N}\exp\left(\gamma_{i,j}\right)}$$
where:$$\gamma_{i,j}=\Psi {\left(x_{i}\right)}^{T}\Omega \left(x_{j}\right).$$

The output of the attention layer is given as:
(3)$$o_{j}=v\left(\sum_{i=1}^{N}\beta_{i,j}h\left(x_{i}\right)\right)$$
where:$$h\left(x_{i}\right)=W_{h}x\;\mathrm{and}\;v\left(x_{i}\right)=W_{v}x$$

Lastly, the final output of the attention module, $y_{i}$, is computed similar to the identity shortcut connection of the ResNet model by adding the scaled output of the attention layer with the original input feature map as follows:(4)$$y_{i}=\lambda o_{i}+x_{i}$$
where λ is a learnable scalar parameter, initialized as 0. According to [15], λ allows the network to progressively develop the features by relying on cues in the local neighborhood while learning to assign weights to non-local features. Hence, SA allows a network to learn both the local features and their dependency across different regions in the image at a small computational cost.

#### 3.2.3. Mish Activation Function

The activation function is key to optimal performance of a deep learning model in two regards: (1) introducing non-linearity so the model can learn complex nonlinear patterns, and (2) aiding gradient flow during backpropagation which can enhance or impede the network from training. Sigmoid and tanh activation functions saturate faster, which has been shown to impede network training. On the other hand, Rectified Linear Unit (ReLu) has been found to enhance training performance by preserving gradient flow. However, ReLu leads to dying neurons, wherein part of the network becomes inactive during learning and inference.

The ReLu activation function has several desirable features; hence, many researchers have proposed different improvements to the original function, such as Leaky ReLu, exponential ReLu, Parameterized ReLu, etc. However, the simplicity and efficacy of ReLu remain unchallenged. Misra [16] demonstrates a new activation that preserves all the desired properties of ReLu and prevents dying neurons. The activation function, called *mish*, was demonstrated to consistently outperform ReLu and other activation functions on CIFAR-10, CIFAR-100, CalTech-256, and ASL datasets which cut across image classification, segmentation, and generation.

Hence, in this paper, the mish activation function is used throughout the network. Mathematically, mish is given as:(5)$$\begin{array}{rr} f\left(x\right) & =x\cdot \tan h\left(\mathrm{softplu}\left(x\right)\right)\\ & =x\cdot \tan h\left(\ln\left(1+e^{x}\right)\right) \end{array}$$

This implies mish combines tanh and softplus activation functions, making it easy to implement. Figure 4 shows that mish is smooth and non-monotonous.

### 3.3. Modified Training Process

The training process in deep learning involves the forward propagation of input through the network. Then, the loss computation compares the model prediction with the ground-truth label. Finally, the backpropagation of loss through the network is used to update the network parameters. While the forward propagation and backprop have been standardized, the choice of loss function and optimization (a function that updates network parameters) can significantly enhance the network performance. In this light, we discuss two modifications: lookahead optimizer and label smoothing.

#### 3.3.1. Lookahead Optimizer

The choice of learning rate (LR) is very germane to the training process of deep learning models. A large LR means the model may not converge to the local minimum, whereas a very small LR will make the training progress slowly. Several studies have sought to lessen the dependency of optimizers on the choice of LR. However, the optimizers have been demonstrated to increasingly depend on hyperparameters such as learning rate, batch size, momentum, weight decay, etc.

The Lookahead Optimizer developed in [17] works as a wrapper function for other optimizers. It consists of two kinds of weights: the slow weight and the fast weight. The fast weight is computed *k*-steps ahead by any optimizer of choice, and then the slow weight is interpolated in the direction of the fast weight. Hence, the lookahead optimizer takes *k*-steps ahead, then backtracks to update the weight. This idea was demonstrated in [17] on different deep learning tasks and datasets; it was shown to outperform the original optimizer consistently.

Whereas optimizers such as stochastic gradient descent (SGD), RMSprop, Adam, etc., are relatively sensitive to the choice of hyperparameters, the lookahead optimizer is more robust. Consequently, it lessens the heavy burden of hyperparameter tuning and selection. In addition, the lookahead optimizer, in conjunction with other optimizers, converges faster than those optimizers alone.

We consider the lookahead optimizer a groundbreaking discovery in the deep learning model. Furthermore, since it is available in most popular deep learning frameworks, we adopted it in this paper.

#### 3.3.2. Label Smoothing

Since a supervisor trains deep learning models in the form of ground-truth labels, a deep learning model is only as good as its label. Hence, deep learning models are prone to adversarial attacks, especially attacks that mildly alter the underlying distribution of an image such that it still looks to humans like it belongs to the right class, but the model misclassifies it [18].

The vulnerability of deep learning models to adversarial attacks stems from their overconfidence in predicting the right class, which originated from overconfidence in labeling the ground-truth classes. This overconfident ground-truth label has resulted in the model overfitting the input distribution, which can be easily exploited in an adversarial attack [18].

Aside from adversarial attacks, medical images are labeled by medical experts who are also prone to error. This human error is especially common in LUS and histopathology images. Hence, label smoothing relaxes hard, “overconfident” class labeling by creating uncertainty around the label.

Let the input space of the image in the dataset be denoted as X,N is the total number of images in the dataset, and Y={1…K} is the label space, K is the number of classes in the dataset. Given the independent, identically distribution assumption on the data space, the samples are generated as $S_{n}=\left\{\left(x_{1},y_{1}\right),\left(x_{2},y_{2}\right),\ldots \left(x_{n},y_{n}\right)\right\}$. The joint probability distribution, $P\left(X,Y\right)=p_{i}$ is the distribution of true label vector y, usually a binary distribution for one-hot encoding scheme in a multiclass problem.
(6)$$p_{i}=\{\begin{array}{ll} 1 & \;\mathrm{if}\;i=y\\ 0 & \;\mathrm{otherwise}\; \end{array}$$

The goal of a model is to learn a parameterized function, ${ℋ}_{\theta }$, that maps X to Y (i.e., ${ℋ}_{\theta }:X\to Y$). To do this, the last fully connected layer is often designed with the number of neurons equaling K, to output the predicted confidence scores, $q_{i}$ given as a softmax operator:
(7)$$q_{i}\left(z_{i}\right)=\frac{\exp\left(z_{i}\right)}{\sum_{j=1}^{K}\exp\left(z_{j}\right)}$$

During training, the model learns to make $q_{i}$ similar to $p_{i}$, which may lead to overfitting and prone to adversarial attack. Label smoothing proposed the removal of a certain magnitude α from the actual class y and uniformly redistributing it to other classes as follows:(8)$$q_{i}=\left(1-\alpha \right)y_{i}+\alpha q_{i}^{\prime }$$
where:
(9)$$q_{i}^{\prime }=\sum_{K=1,K\neq y_{i}}^{K}y^{\left(K\right)}=\frac{1}{K-1}\left(1-y_{i}\right)$$

The idea of label smoothing was introduced in Inception-v2 [19] but popularized by Goibert and Dohmatob in [18], providing a generalized framework for label smoothing (Equation (8)). Other forms of label smoothing (Equation (9)) introduced by [18] include Boltzmann and second-best label smoothing.

## 4. Experiments

Following the methodology described in Section 3, we designed experiments to investigate the percentage improvement of each of these techniques in the problem domain. Thus, five different models were trained in different combinations of the techniques described in Section 3.2 and Section 3.3. These experiments were carried out using python fastai deep learning framework on Google Colab (pro version) environment with Nvidia Tesla T4 16GB GPU. The parameters for all experiment modes are presented in Table 3.

The goal of these experiments was to diagnose COVID patients using different medical image modalities, as shown in Figure 1. In Approach 1, we trained a ResNet model first to classify the images into their respective modality, and then the images were passed to our pre-trained models developed in [20] for CXR, [21] for CT scan, and [10] for LUS. We discovered that the overall performance of the model does not improve the results obtained in the previous works cited. However, the ResNet model can degrade the performance if a wrong class is predicted. We concluded that there is no sufficient improvement in this regard and that the ResNet model created an unnecessary computational overhead and performance bottleneck. Hence, we focused on Approach 2 (see Figure 1b).

Approach 2 aims to develop a single model to diagnose COVID-19 patients using different image modalities. We experimented with 5 different 50-layer deep CNN models. (1) A base ResNet model with weight initialized using the Glorot uniform weight initialization method [22] was trained from scratch with parameters, as shown in Table 3. (2) The ResNet model was then fine-tuned using the discriminative fine-tuning methods we introduced in [20]. (3) The XResNet model introduced in [13] was also trained using the pre-trained weight of [13]. (4) Then, the XResNet model was modified by replacing the residual modules with self-attention modules as discussed in Section 3.2.2; this network was trained from scratch with weights initialized using the Glorot uniform weight initialization method [22]. (5) Finally, XResNet with self-attention was also trained from scratch, using the lookahead optimizer discussed in Section 3.3.1 for weight updates. All the training parameters of each network are presented in Table 3.

## 5. Results

The results of the experiments are presented and discussed in this section. We start by investigating the model improvements as a result of the techniques discussed in Section 3.2 and Section 3.3. With all models trained on the same dataset, batch size, and the number of epochs on the same GPU machine, we observed that the average time to complete one epoch is the same for all models (epoch time = 2 min 25 s). Hence, there is no gain or loss of training time. We then compared their training loss, as plotted in Figure 5.

From the graph, XResNet with self-attention trained with Adam optimizer incurred the highest loss, but the loss was drastically reduced by wrapping the lookahead optimizer around. The training loss of XResNet with self-attention at the 20th epoch was 0.544629, which was reduced to 0.432076 by lookahead, yielding a 79% reduction at the 20th epoch. Discriminative Fine-tuning (DFT) is a powerful transfer learning approach introduced in our previous work [20]. It achieves faster convergence and optimal performance by allowing each network layer to learn at its own pace (i.e., assigning different learning rates to each network layer). As shown in Figure 5, DFT achieves minimum training loss.

In addition, from Figure 5, the ResNet model appears to perform better than all the additional techniques we introduced. A further investigation is in place. The training accuracy of each model is presented in Table 4. Recall that ResNet, XResNet with self-attention (XresNet + SA), and XResNet with self-attention and lookahead (XresNet + SA + LA) models were trained from scratch, while XResNet and ResNet with DFT were fine-tuned using transfer learning. From the table, ResNet outperforms XResNet+SA because the self-attention module is quite sensitive to hyperparameters (especially the learning rate). By implementing a lookahead optimizer, the self-attention module becomes more robust and insensitive to hyperparameters, improving the result obtained. Note that the XResNet + SA + LA model’s performance is a relatively good result for a model trained from scratch and that the fine-tuned XResNet only gained a +1.02 improvement over it. Similarly, ResNet with DFT performs better than ResNet only, gaining a +10.21 improvement over ResNet by achieving an accuracy of 91.26%.

The accuracy obtained by the models reflects the complexity of training a single model to classify COVID-19 from multiple medical imaging modalities. We present the confusion matrix of each model in Figure 6 to investigate how each model discriminates and classifies the images. Note that a raw image input is supplied, and the model needs to predict the right image modality correctly and the infection without being explicitly told (see how the classes are labeled in Figure 6). The diagonal of the matrices represent areas where the models correctly predict the correct image modality and disease infection, while the off-diagonal represents misclassifications.

Considering Figure 6, along each model’s accuracy as presented in Table 4, models with higher accuracy first learn to correctly predict the right image modality and then diagnose the disease. The models in Figure 6b,d,e correctly predicted the image modalities but misdiagnosed a few examples (classifying COVID as non-COVID and vice versa). This supports our hypothesis that a single model can diagnose multi-imaging modality and that to obtain high accuracy, the model will first learn to discriminate each imaging modality before predicting the disease.

Lastly, the class-specific sensitivity and specificity of each model are calculated using Equations (10) and (11), respectively. In a multiclass classification, the sensitivity of a model is the ability of the model to predict a particular class correctly. In contrast, sensitivity is the ability of the model to correctly predict that an image does not belong to a particular class [23].
(10)$$MSN_{i}=\frac{TP_{i}}{TP_{i}+FN_{i}}$$
(11)$$MSP_{i}=\frac{TN_{i}}{TN_{i}+FP_{i}}$$
where:
(12)$$\begin{array}{c} TP_{i}={\left.C_{ij}\right|}_{i=j}\\ FP_{i}=\sum_{i}C_{ij}-TP_{i}\\ FN_{i}=\sum_{j}C_{ij}-TP_{i}\\ TN_{i}=\sum_{i}\sum_{j}C_{ij}-TP_{i}-FN_{i}-FP_{i} \end{array}$$
where $C_{ij}$ represents the confusion matrix of the model, with row i and column j. $TP_{i}$ is the true positive rate for class i which measures the number of images correctly classified as class i, whereas $TN_{i}$ is the true negative rate for class i which measures the number of images that are rightly classified as a non-member of class i. Conversely, $FP_{i}$ is the false positive rate of class i which quantifies the total number of images that are wrongly predicted to belong to class i, whereas $FN_{i}$ is the false negative rate which gives the total number of images that belong to class i but the model predicted that it belongs to another class.

Using these formulae, the class-specific sensitivity and specificity of each model were calculated. The result is presented in Figure 7.

## 6. Conclusions

Medical imaging presents a difficult task for deep learning models. The model is required to discover subtle changes in images that represent landmarks and features used by a human radiologist in diagnosing a particular ailment. These features are sometimes microscopic, like a pin in a haystack, in which the feature of interest is covered by numerous unwanted information. In addition, the field of medical diagnosis requires high precision and sensitivity, as the cost of misdiagnosis can be very high. These twin challenges are complicated when training a model using multiple modalities; as a result, few research works are relevant.

This paper presents an improved end-to-end method of multimode image classification using deep learning. We presented top research methods developed over the years to improve models trained from scratch and transfer learning approach. We trained three models from scratch and used various methods to improve the model’s training loss, accuracy, sensitivity, and specificity. We showed that when fully trained, a model can first discriminate the imaging modality by itself and then diagnose the relevant disease. The model that achieved the highest accuracy correctly maps all input images to their respective modality and then classifies the disease (with some errors).

In future work, we seek to investigate the class activation map of each model using explainable AI techniques with excellent visualization. This will enhance our understanding of how those models handle each imaging modality, what portion of the image they consider in deciding, and how that differs for each imaging mode.

## Figures and Tables

**Figure 1 bioengineering-09-00312-f001:**
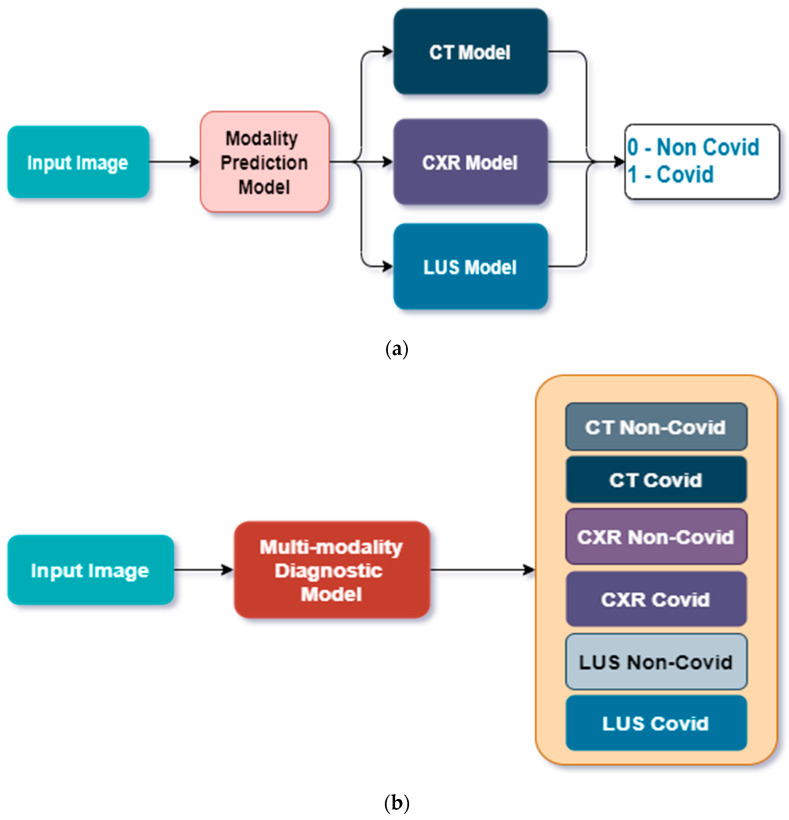
General overview of the methodology. (**a**) Approach 1: A CNN predicts the imaging modality and then passes the image to one of three special CNN models designed for each image mode to predict COVID or Non-COVID diagnosis, (**b**) Approach 2: A single CNN model predicts both image modality and the correct class for each input image.

**Figure 2 bioengineering-09-00312-f002:**
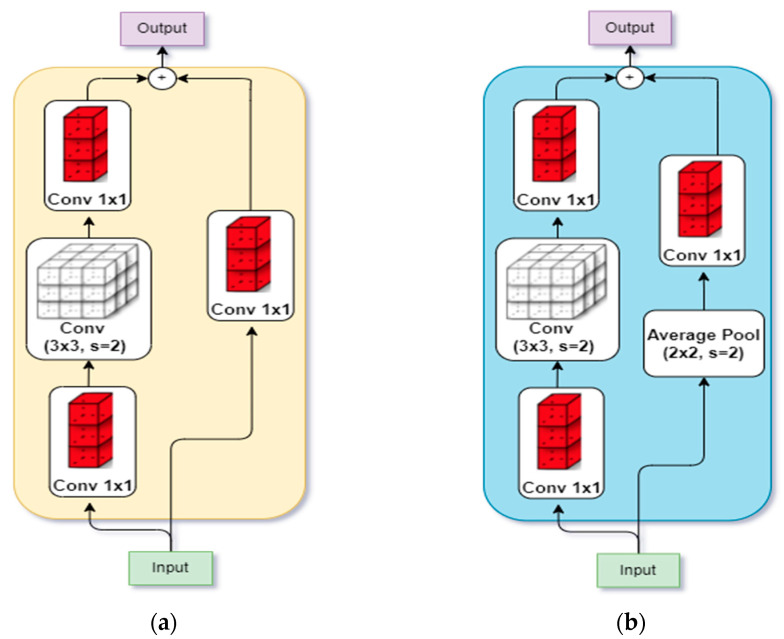
Architectural Modification by [13]. (**a**) Down-sampling in original ResNet [12]; (**b**) Down-sampling in XResnet [13].

**Figure 3 bioengineering-09-00312-f003:**
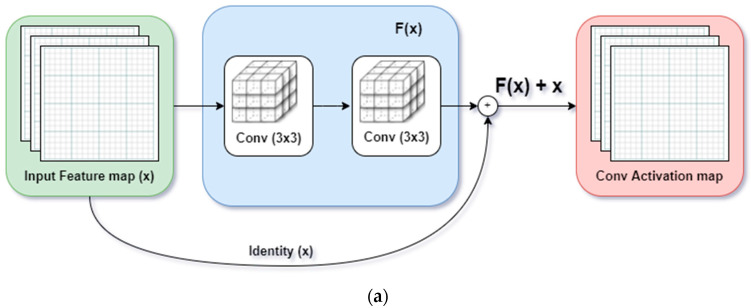
Proposed ResNet-Self Attention model: The Residual Block is replaced with Self-Attention Module. (**a**) Residual Block implemented with identity mapping by shortcut in [12]; (**b**) Self-Attention Module proposed in [15].

**Figure 4 bioengineering-09-00312-f004:**
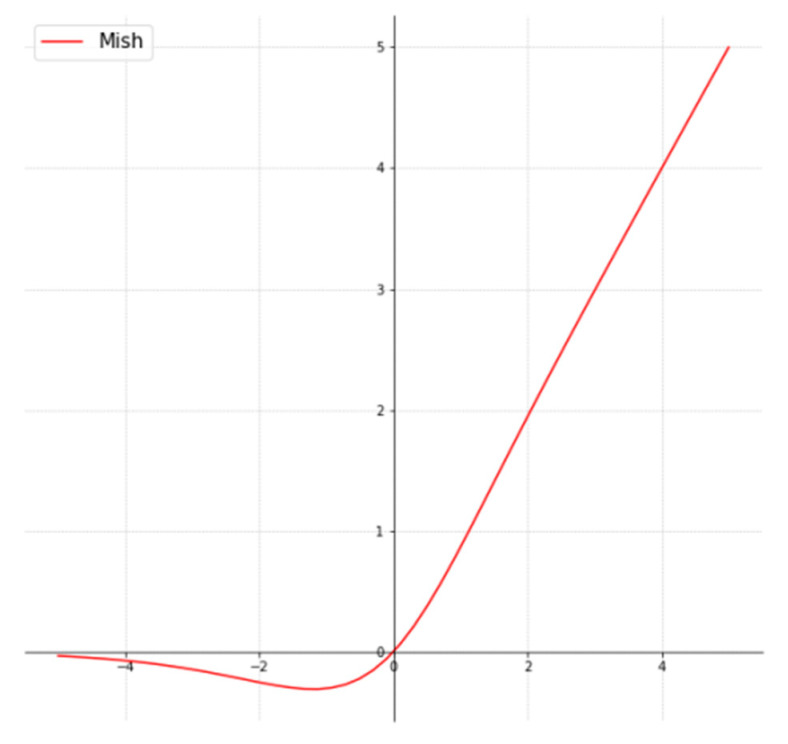
Mish Activation Function [16].

**Figure 5 bioengineering-09-00312-f005:**
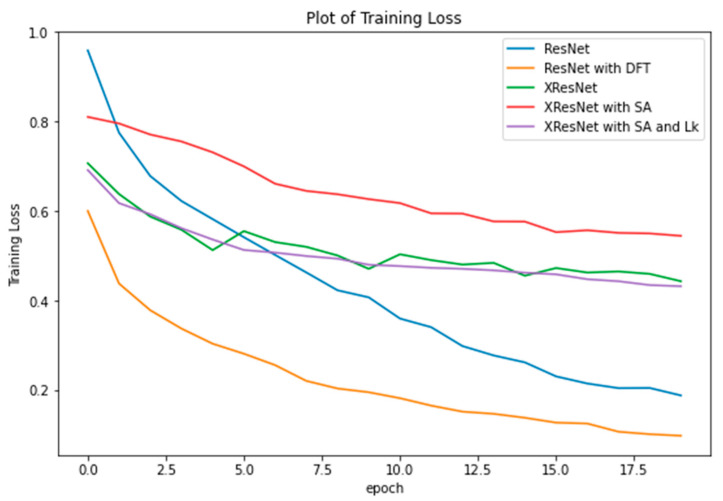
Training Loss of ResNet trained from scratch, ResNet fine-tuned using DFT, XResNet initialized with pre-trained weights, XResNet with self-attention trained from scratch, and XResNet with self-attention and lookahead optimizer.

**Figure 6 bioengineering-09-00312-f006:**
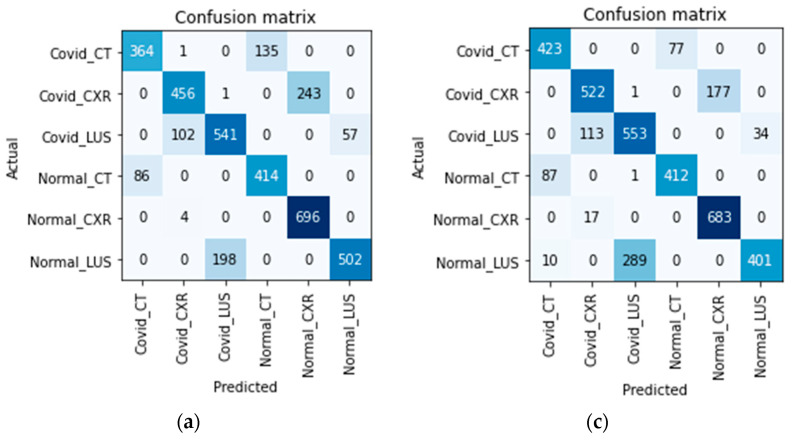
The confusion matrix of the models on validation dataset. (**a**) Confusion matrix of ResNet Model; (**b**) Confusion matrix of XResNet only; (**c**) Confusion matrix of XResNet with Self-Attention; (**d**) Confusion matrix of XResNet + Self-Attention + Lookahead Optimizer; (**e**) Confusion matrix of ResNet with DFT.

**Figure 7 bioengineering-09-00312-f007:**
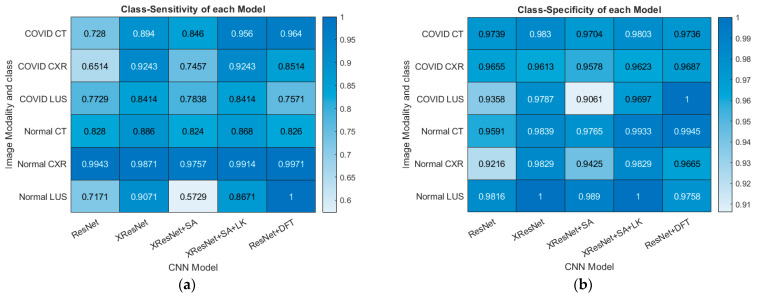
Performance Metrics Statistics on the Validation dataset. (**a**) Class-specific Sensitivity for each Model (**b**) Class-specific Specificity for each Model.

**Table 1 bioengineering-09-00312-t001:** Distribution of images in the dataset.

Image Modality	No. of COVID	No. of Non-COVID
Lung Ultrasound	60 Videos	56 Videos
CT Scan	1252 Images	1230 Images
Chest X-ray	3616 Images	10,192 Images

**Table 2 bioengineering-09-00312-t002:** Data Augmentation Transformation.

Data Augmentation	Parameter	Value
Random Gaussian Blurring	Kernel size	3
Random zooming	Scale	1.3
Random Affine	Magnitude	0.4
Random lighting	Intensity	1.4

**Table 3 bioengineering-09-00312-t003:** Training Parameters for each model.

Parameter	ResNet	XR *	XR + SA *	XR + SA *	ResNet + DFT *
Image Size	224	224	224	244	224
Batch Size	64	64	64	64	64
# of epochs	20	20	20	20	20
Optimizer	Adam	Adam	Adam	LK * + Adam	Adam
Act. Func.	ReLu	Mish	Mish	Mish	ReLu
Loss Func.	CE *	LS *	LS *	LS *	CE *
Learning rate	0.001	0.002	0.006	0.0006	0.001
Momentum	0.99	0.99	0.99	0.99	0.99
Weight decay	0.01	0.01	0.01	0.01	0.01

* XR: XResNet, XR + SA: XResNet with Self-attention, DFT: Discriminative Fine-tuning, LK: look-ahead optimizer, CE: crossentropy loss, LS: label smoothing.

**Table 4 bioengineering-09-00312-t004:** Training Accuracy for each model.

Model Middle Line Is 0.5pt	Accuracy (%)
ResNet	80.05
XResNet	91.07
XResNet with Self Attention	72.47
XResNet with Self Attention and Lookahead	90.05
ResNet with DFT	91.26

## Data Availability

All the datasets are available online and have been duly referenced within the text.

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
