# Peer review of "Bag of Tricks for Improving Deep Learning Performance on Multimodal Image Classification"

_bioengineering, 2022, doi:10.3390/bioengineering9070312_

Round 1

Reviewer 1 Report

The authors describe a method by which multimodal image classification using machine learning methods, can be used to classify disease with above 90% accuracy. The study is relevant from 2 important perspectives: firstly, the rapid development of artificial intelligence in extrapolating information from large datasets, and secondly, the major impact of SARS-CoV2 globally on health, economics and lifestyle. The text is well-written and relevant methods and figures included. However, various typographical, formatting and linguistic errors need to be addressed (see attached document)

Author Response

We thank our dear reviewer for the ample time dedicated to reviewing this work. We value all your comments, corrections and recommendations and have carefully considered and implemented them.

Kindly find our response attached.

Thank you, warm regards.

Reviewer 2 Report

Development of software approaches for classification of medical images has a potential to streamline and improve medical diagnostics, and is therefore of great importance. In the last years, great progress has been made in the development of deep learning methods for image classification. In this manuscript, the authors investigated the applicability of relevant deep learning methods for diagnosing COVID-19. They evaluated some of the recently developed deep neural net architectures and optimization techniques, for classification of X-ray, CT scan and ultrasound images. The manuscript addresses a relevant issue and is well written. While it appears that the results are sound, the omission of crucial data precludes a full assessment of the manuscript (points 1 and 2). Here are specific issues:
  1. (Line 354 and below) The authors state that training loss is shown in Figure 1. However, Figure 1 shows a general overview of the methodology.
  2. (Line 387-392) Figure 2 is also supposed to show the results, but instead it shows ResNet architecture.
  3. Some numbers are hard to read on Figure 6
  4. The resolution of Figure7 should be improved and the text box that covers almost the entire two bottom lines should be removed
  5. (Figure 7) It is confusing to report sensitivity values for Normal CT/CXR/LUS because sensitivity is defined using TPs and only the three values where COVID-19 was predicted in the actual COVID-19 case are TPs. For example, the value 364 in Figure 6a is a TP (used correctly to calculate sensitivity of 0.728 in Figure 7a, ResNet column), while 414 (Figure 6a) is TN so the value of 0.828 (Figure 7a, ResNet column) does not represent sensitivity. The analogous argument applies to specificity. In addition, it would be useful to state how exactly were sensitivity and selectivity calculated from the data presented in tables of Figure 6.
  6. At some places in the text COVID19 appears without “19”
  7. (Line 44) Section numbers should be corrected
  8. (Line 87) “MGA” instead of “GA”
  9. (Line 146) Please specify the “various transformations”
  10. (Figures 2 and 3) Shouldn't the convolution and the shortcut paths be added (as in Eq 1) and not subtracted as the “-” sign indicates?
  11. (Lines 285-286) “Since” and “consequently” should not be used together.
  12. (Lines 428-435) Real initials should be added
  13. Several references are incomplete

Author Response

We thank our esteemed reviewer for the ample time dedicated to reviewing this work. We cherish your comments, corrections and recommendations and have carefully attended to and implemented them.

We hope our efforts are considered sufficient by our esteemed reviewer. Kindly find our response attached.

Thank you, warm regards.

Round 2

Reviewer 2 Report

The authors adequately addressed most of my concerns from the previous round. There are only few minor points:
  1. Line 186: The strides for the second and the third convolution filter (3x3 and 1x1) are not specified
  2. Line 207: “In a bit to” should be changed
  3. Line 223: It is not clear what is “sequence”
  4. Section 3.2.1: Please specify the depth of all convolution layers
  5. Figure 3b: It would be useful to add psi, omega and beta (or gamma) on the figure
  6. Eq2: Beta is defined but never used
  7. Line 397: Please add citation for “our previous work” and explain briefly how DFT works
  8. Line 428: Please explain how is the temporal sequence “first learn ... then diagnose ...” obtained from the data shown on Figure 6.
  9. Figure 6: It would be easier to explain the previous point if the order of rows/columns was: Covid CT, Normal CT, Covid CXR, Normal CXR, Covid LUS, Normal LUS
  10. Lines 450, 451: The equations are not correct because the right hand sides of these two equations are the same
  11. Lines 450, 451: The equations are not correct because there is a problem with summation indices (j is not specified)
  12. Figure 7: Some values do not appear to be calculated correctly. For example, taking ResNet with DFT (Figure 6e), sensitivity of Covid CT should be: 482 / (482 + 18) = 0.964, while the corresponding value in Figure 7a is 0.95. Similarly, for Covid LUS, sensitivity should be: 530 / (530 + 95 + 75) = 0.76, wile it is 0.7029 in Figure 7a.
  13. Figure 7. It appears that the authors did not understand the point 5 from my previous comments. The authors used the formulas for multi-classification sensitivity and specificity as if there were six independent classes, that is without taking into account that in their data there are three positive (Covid CT, Covid CXR and Covid LUS) cases and three negative (the three Normal cases). Therefore, TPs and sensitivity can be calculated only for the three positive cases. The analogous argument can be made for specificity. I will not make further objections if the authors insist to keep the current way of calculating sensitivity and specificity, but they should state this point in the manuscript.
  14. Lines 454, 459: Please specify what are C_ij, MSN_i and MSP_i

Author Response

We appreciate our esteemed reviewer for the quality time dedicated to reviewing this work. Your comments and corrections are genuinely appreciated, they have helped improve the original manuscript.

At the moment, we have carefully attended to all your comments and correction. Kindly find the attached response.

Warm regards.
